# Dynamic Linear Attention

Xin Wang [* 1]   Hui Shen [* 2]   Boyuan Zheng [2]   Xueshen Liu [2]   Minkyoung Cho [2]   Zhongwei Wan [1]   Zesen Zhao [2]
Zhuoqing Mao [2]   Shen Yan [3]   Mi Zhang [1]

## Abstract

The scalability of Large Language Models (LLMs) to long contexts is fundamentally constrained by the quadratic complexity of standard attention, motivating the adoption of linear attention mechanisms with sub-quadratic cost. To improve representation capacity under long contexts, recent approaches organize memory in a multi-state manner. However, existing multi-state linear attention methods rely on fixed state merging policies that cannot adapt to dynamically varying token importance, irreversibly obscuring critical tokens and causing severe error accumulation over long sequences. To address this limitation, we propose `DLA`, a dynamic memory modeling framework for multi-state linear attention. `DLA` introduces (i) *Information-Aware Dynamic State Merging*, which adaptively determines state boundaries based on token-level information variation, preserving high-resolution representations around semantic transitions while aggressively summarizing stable regions, and (ii) *Capacity-Bounded Memory Modeling*, which maintains a fixed-size, chronologically ordered state cache by selectively merging adjacent low-information states to control memory growth with minimal information loss. We pre-train `DLA` on two different linear attention models and evaluate on 16 datasets across three categories. Experimental results demonstrate the superiority of `DLA` over state-of-the-art.

## 1. Introduction

Large Language Models (LLMs) have demonstrated remarkable capabilities across a wide range of natural language understanding and generation tasks. However, scaling LLMs to long-context settings remains a fundamental challenge due to the quadratic computational and memory complexity of standard self-attention (Wan et al., 2023; Wang et al., 2024; Wan et al., 2025). This limitation has motivated extensive research on efficient attention mechanisms that enable long-sequence modeling without retraining from scratch. Among these approaches, linear attention (Yang et al., 2024; 2025) has emerged as a promising direction, as it approximates full attention with sub-quadratic complexity and offers favorable scalability to long contexts.

To further improve the representation capacity of linear attention under long sequences, recent works organize historical context in a multi-state manner, where long token histories are partitioned into chunks and summarized into compact memory states. Representative methods such as Log-Linear Attention (Guo et al., 2025) demonstrate improved efficiency and practicality for long-context inference. By operating on summarized states rather than individual tokens, these approaches significantly reduce memory footprint and computation cost.

Despite their success, existing multi-state linear attention methods still suffer from notable performance degradation as context length increases. This limitation stems from a fundamental mismatch between fixed memory construction policies and the non-uniform, dynamically evolving information structure of long sequences. In particular, current methods typically rely on fixed block sizes or rule-based merging schedules, implicitly assuming uniform information density across the sequence. Such designs fail to adapt to dynamically emerging semantic transitions, forcing critical tokens to be prematurely absorbed into coarse summaries. Moreover, merge decisions made under fixed policies are irreversible: once heterogeneous tokens are compressed into a single state, their individual contributions cannot be recovered, leading to error accumulation.

These observations suggest that effective long-context linear attention requires memory modeling mechanisms that are both information-aware and capacity-controlled. On one hand, state construction should adapt to local representation variation, allocating higher resolution to semantically volatile regions while aggressively summarizing stable spans. On the other hand, the total number of memory states

---

[*]Equal contribution  [1]The Ohio State University  [2]University of Michigan  [3]Bytedance Seed. Correspondence to: Xin Wang <wang.15980@osu.edu>, Mi Zhang <mizhang.1@osu.edu>.

must be explicitly bounded to ensure predictable computation and memory cost during inference.

In this work, we propose Dynamic Linear Attention (DLA), a new framework for multi-state linear attention that addresses these challenges. DLA differs from prior approaches in two key aspects. First, DLA introduces Information-Aware Dynamic State Merging, which determines state boundaries on the fly based on token-level information variation. Instead of relying on fixed merging policies, DLA evaluates the representation change of each incoming token relative to the current memory state, merging low-variation tokens while initiating new states at semantic transition points. Second, DLA incorporates Capacity-Bounded Memory Modeling, which maintains a fixed-size, chronologically ordered state cache. When the cache reaches its capacity, DLA selectively merges adjacent low-information states, preserving temporal order while minimizing information loss.

We pre-train DLA on two linear-attention backbones, Mamba-2-780M and Gated DeltaNet-1.3B, following the design in (Guo et al., 2025). We evaluate DLA on 16 datasets spanning three aspects: eight commonsense reasoning benchmarks, six in-context retrieval datasets, and two long-context modeling datasets. We highlight three main findings. (1) DLA consistently outperforms the state-of-the-art multi-state method, Log-Linear Attention, across all tasks. (2) When applied to Mamba-2, the DLA variant even achieves performance comparable to full-attention Transformers with similar parameter budgets. (3) DLA achieves superior efficiency, delivering higher throughput and lower runtime memory consumption than Log-Linear Attention.

## 2. Preliminary

We consider a sequence modeling task with input length $T$ and hidden dimension $d$. Let $\mathbf{Q}, \mathbf{K}, \mathbf{V} \in \mathbb{R}^{T \times d}$ denote the query, key, and value matrices. Standard self-attention computes the output $\mathbf{O} \in \mathbb{R}^{T \times d}$ as

$$\mathbf{O} = \mathrm{softmax}(\mathbf{Q}\mathbf{K}^\top \odot \mathbf{M})\mathbf{V},$$

where $\mathbf{M}$ is the causal mask. While effective, this operation incurs quadratic time and memory complexity in $T$, motivating the development of sub-quadratic attention mechanisms. In this section, we review linear attention and its multi-state variants that form the foundation of our approach.

### 2.1. Linear Attention

Linear attention mitigates the quadratic cost of Transformers by removing the softmax normalization, enabling the reordering of computation via associativity. A causal linear attention layer can be written in a parallel form as

$$\mathbf{O} = (\mathbf{Q}\mathbf{K}^\top \odot \mathbf{M})\mathbf{V}, \qquad \mathbf{M}_{ij} = \mathbb{I}(i \geq j). \quad (1)$$

This formulation admits an equivalent recurrent implementation. Let $\mathbf{q}_t, \mathbf{k}_t, \mathbf{v}_t \in \mathbb{R}^d$ denote the query, key, and value vectors at time step $t$. Linear attention maintains a single state matrix $\mathbf{S}_t \in \mathbb{R}^{d \times d}$ that summarizes all past tokens:

$$\mathbf{S}_t = \mathbf{S}_{t-1} + \mathbf{v}_t\mathbf{k}_t^\top, \quad (2)$$

$$\mathbf{o}_t = \mathbf{S}_t\mathbf{q}_t. \quad (3)$$

This recurrent form enables linear-time inference with constant memory, but compresses the entire history into a single state, which can limit representation capacity under long contexts. We use $\phi(\cdot)$ to denote the feature map used in linear attention. Unless otherwise specified, $\phi : \mathbb{R}^d \to \mathbb{R}^d$ is implemented as an identity mapping or a learnable linear projection, following prior work.

### 2.2. Linear Attention with the Delta Rule

To improve state tracking and introduce controlled forgetting, DeltaNet (Yang et al., 2024) extends linear attention with a delta-style update rule:

$$\mathbf{S}_t = \mathbf{S}_{t-1}(\mathbf{I} - \beta_t\mathbf{k}_t\mathbf{k}_t^\top) + \mathbf{v}_t\mathbf{k}_t^\top, \quad (4)$$

where $\beta_t$ is a data-dependent step size. While this formulation improves adaptivity over a pure accumulator, it still relies on a single global state and therefore cannot selectively preserve fine-grained information over long sequences.

### 2.3. Multi-State Linear Attention

To increase modeling capacity while retaining sub-quadratic complexity, recent work organizes linear attention in a multi-state manner by partitioning the historical context into segments and summarizing each segment into a separate state (Guo et al., 2025; Wei et al., 2025). Among them, log-linear attention (Guo et al., 2025) replaces the single recurrent state with a logarithmic number of multi-scale states constructed via a Fenwick-tree decomposition of the causal prefix. Concretely, at time step $t$, the prefix $[0, t]$ is decomposed into at most $L = \lceil \log_2(t+1) \rceil + 1$ disjoint buckets $\{B_t^{(\ell)}\}_{\ell=0}^{L-1}$, with finer resolution near the current position and coarser resolution for distant history. The corresponding linear attention states and the final aggregated output are computed as:

$$\mathbf{S}_t^{(\ell)} = \sum_{s \in B_t^{(\ell)}} \mathbf{v}_s\mathbf{k}_s^\top \in \mathbb{R}^{d \times d}, \mathbf{o}_t = \sum_{\ell=0}^{L-1} \lambda_t^{(\ell)}\, \mathbf{S}_t^{(\ell)}\mathbf{q}_t \quad (5)$$

This design achieves $O(T \log T)$ training complexity and $O(\log T)$ time and memory per decoding step. However, the granularity of its memory states is determined by a fixed hierarchical schedule, independent of token-level representation variation. As a result, semantically salient tokens may be prematurely absorbed into coarse summaries, and

**Algorithm 1** Information-Aware Dynamic State Merging

1: **Input:** Token States $\{s_t\}_{t=1}^T$
2: **Output:** memory states $\mathcal{M} = \{S_i\}$
3: $\mathcal{M} \leftarrow [\,]$
4: **for** $t = 1$ **to** $T$ **do**
5:    **if** $\mathcal{M}$ is empty **then**
6:       $\mathcal{M} \leftarrow \{s_t\}$; **continue**
7:    **end if**
8:    $S \leftarrow$ last state in $\mathcal{M}$
9:    $I_t \leftarrow \frac{\|s_t - S\|_F}{\|S\|_F + \epsilon}$
10:   replace last state in $\mathcal{M}$ with $\mathrm{Merge}(S, s_t)$
11:   **if** $I_t \geq \tau$ **then**
12:      append $s_t$ to $\mathcal{M}$;
13:   **end if**
14: **end for**
15: **return** $\mathcal{M}$

disturbances introduced at critical positions can propagate through the fixed multi-scale states. This limitation motivates the need for information-aware and adaptive memory construction, which we address in the next section.

# 3. Dynamic Linear Attention (DLA)

Figure 1 provides an overview of DLA. DLA is an information-aware linear attention framework that dynamically constructs a compact set of memory states for efficient long-context modeling. Unlike prior approaches that rely on fixed temporal schedules or predefined block boundaries, DLA adaptively determines state granularity based on token-level information variation. Specifically, tokens are processed sequentially. For each new token, DLA computes a lightweight *State Information Score* measuring its representation change relative to the most recent memory state. Tokens with low information variation are merged into the current state, while tokens exhibiting significant drift initiate a new state. This enables fine-grained modeling around semantic transitions while aggressively summarizing stable token spans. To bound memory and computation, DLA maintains a capacity-limited state cache. When the cache reaches its maximum size, two adjacent states with the lowest information density are merged, preserving temporal order while minimizing information loss. The resulting memory consists of a fixed-size, chronologically ordered set of summary states. At each decoding step, DLA produces the output by attending over the maintained memory states using a linear attention formulation, where each state contributes with a query-dependent weight. Together, information-aware state construction and capacity-bounded memory modeling enable DLA to achieve adaptive resolution, stable inference cost, and efficient long-context representation.

## 3.1. Information-Aware Dynamic State Merging

**Motivation:** Existing multi-block linear attention methods typically rely on fixed schedules (e.g., block and merge every $K$ tokens) (Guo et al., 2025) or hard, rule-based boundaries (Wei et al., 2025) to determine the block of historical tokens that should be merged into summary states. While such designs improve memory and compute efficiency, they are largely agnostic to the semantic evolution of the sequence. In practice, information density is highly non-uniform: critical semantic transitions may occur abruptly, whereas long stretches of tokens can be locally redundant. As a result, fixed or hard block policies often suffer from **two key limitations**. First, they cannot adapt to dynamically emerging semantic changes, forcing important transitions to be prematurely absorbed into coarse summaries simply because a pre-defined boundary is reached. Second, merge decisions made without regard to local semantic continuity are inherently irreversible: once tokens are merged under a fixed policy, their individual contributions cannot be recovered, even if subsequent context reveals their importance. These misalignments between merge decisions and the true semantic structure lead to sub-optimal generation and finally degrades the representation quality.

In the following, we provide a theoretical proof on why the fixed merging policy is sub-optimal.

**Theorem 3.1** (State deviation). *Let $\{u_t\}_{t=1}^T \subset \mathbb{R}^d$ denote per-token additive contributions to a linear attention state. Consider a blocking policy $\pi$ of token list $\{1, \ldots, T\}$ into $m$ disjoint contiguous blocks $\{\mathcal{C}_i\}_{i=1}^m$. For each block $\mathcal{C}_i$, let $\bar{u}_i \in \mathbb{R}^d$ be a representative summary vector. Therefore, for any query vector $q \in \mathbb{R}^d$, the exact output $y(q)$ and the summarized output $\tilde{y}_\pi(q)$ for a query vector are:*

$$y(q) \triangleq \sum_{t=1}^T \langle q, u_t \rangle, \quad \tilde{y}_\pi(q) \triangleq \sum_{i=1}^m \sum_{t \in \mathcal{C}_i} \langle q, \bar{u}_i \rangle \quad (6)$$

*The deviation induced by summarization* $\mathrm{Err}(\pi; q) \triangleq |y(q) - \tilde{y}_\pi(q)|$ *admits the bound:*

$$\left| y(q) - \tilde{y}_\pi(q) \right| \leq \|q\|_2 \cdot \sum_{i=1}^m \sqrt{|\mathcal{C}_i|} \sqrt{\sum_{t \in \mathcal{C}_i} \|u_t - \bar{u}_i\|_2^2} \quad (7)$$

*Proof.* The deviation between the exact and summarized outputs $y(q) - \tilde{y}(q)$ can be further rewritten as:

$$\sum_{i=1}^m \sum_{t \in \mathcal{C}_i} \langle q, u_t - \bar{u}_i \rangle = \left\langle q, \sum_{i=1}^m \sum_{t \in \mathcal{C}_i} (u_t - \bar{u}_i) \right\rangle \quad (8)$$

By Applying Cauchy–Schwarz (Johnston et al., 2025) In-

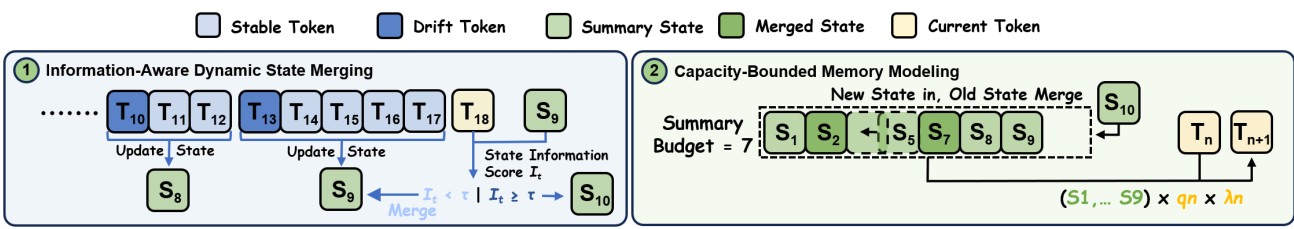

Figure 1. Overview of DLA.

equality, we have:

$$\text{Err}(\pi; q) = |y(q) - \tilde{y}_\pi(q)| \leq \|q\|_2 \cdot \left\| \sum_{i=1}^{m} \sum_{t \in \mathcal{C}_i} (u_t - \bar{u}_i) \right\|_2 \tag{9}$$

We then use Triangle Inequality (Lumezanu et al., 2009) over chunks to further get:

$$\left\| \sum_{i=1}^{m} \sum_{t \in \mathcal{C}_i} (u_t - \bar{u}_i) \right\|_2 \leq \sum_{i=1}^{m} \left\| \sum_{t \in \mathcal{C}_i} (u_t - \bar{u}_i) \right\|_2 \tag{10}$$

Similarly, for each block $\mathcal{C}_i$, we also have:

$$\left\| \sum_{t \in \mathcal{C}_i} (u_t - \bar{u}_i) \right\|_2 \leq \sum_{t \in \mathcal{C}_i} \|u_t - \bar{u}_i\|_2$$

$$\leq \sqrt{|\mathcal{C}_i|} \cdot \sqrt{\sum_{t \in \mathcal{C}_i} \|u_t - \bar{u}_i\|_2^2}. \tag{11}$$

By combining (9), (10), and (11), we finally obtain the upper-bound $B(\pi; q)$ of the deviation $\text{Err}(\pi; q)$:

$$B(\pi; q) \triangleq \|q\|_2 \sum_{i=1}^{m} \sqrt{|\mathcal{C}_i|} \sqrt{\sum_{t \in \mathcal{C}_i} \|u_t - \bar{u}_i\|_2^2} \tag{12}$$

This upper bound shows that the deviation induced by block-wise summarization is controlled by the within-block heterogeneity. As a result, content-agnostic fixed blocking policies, which do not adapt to representation variation, can incur a larger bound on non-stationary sequences, especially when tokens with large representation variance are mixed into the same block. □

**Corollary 3.2** (Fixed blocking is sub-optimal on non-stationary sequences). *There exists a class of non-stationary token sequences for which any fixed blocking policy $\pi_{\text{fix}}$ yields a strictly larger deviation bound $B(\pi_{\text{fix}}; q)$ than an adaptive blocking policy $\pi_{\text{dyn}}$ that aligns block boundaries with semantic change points.*

*Proof sketch.* Consider a non-stationary sequence consisting of two contiguous segments $\mathcal{A}$ and $\mathcal{B}$ with distinct means $\mu_A \neq \mu_B$. For simplicity, assume $u_t = \mu_A$ for $t \in \mathcal{A}$ and $u_t = \mu_B$ for $t \in \mathcal{B}$ (a special case of $u_t \sim \mathcal{D}_A, \mathcal{D}_B$).

Let $\pi_{\text{fix}}$ be any fixed blocking policy that yields at least one block $\mathcal{C}$ overlapping both segments, and denote $n_A = |\mathcal{C} \cap \mathcal{A}|$, $n_B = |\mathcal{C} \cap \mathcal{B}|$. For this block, the choice $\bar{u}$ that minimizes $\sum_{t \in \mathcal{C}} \|u_t - \bar{u}\|_2^2$ is the block mean $\bar{u} = \frac{n_A \mu_A + n_B \mu_B}{n_A + n_B}$, and the minimum within-block heterogeneity satisfies

$$\sum_{t \in \mathcal{C}} \|u_t - \bar{u}\|_2^2 = \frac{n_A n_B}{n_A + n_B} \|\mu_A - \mu_B\|_2^2 > 0.$$

In contrast, an adaptive policy $\pi_{\text{dyn}}$ that places a boundary at the change point produces blocks contained in $\mathcal{A}$ or $\mathcal{B}$ only, for which the optimal heterogeneity term is 0 under the same construction. Since the deviation bound $B(\pi; q)$ is a sum of nonnegative per-block terms $\|q\|_2 \sqrt{|\mathcal{C}_i|} \sqrt{\sum_{t \in \mathcal{C}_i} \|u_t - \bar{u}_i\|_2^2}$, the overlapping block $\mathcal{C}$ alone contributes a strictly positive amount to $B(\pi_{\text{fix}}; q)$ for any $q \neq 0$, while $B(\pi_{\text{dyn}}; q)$ does not incur this cross-segment term. Hence, there exists such a sequence for which $B(\pi_{\text{fix}}; q) > B(\pi_{\text{dyn}}; q)$, proving the claim. □

**Key Design:** The pseudocode of Information-Aware Dynamic State Merging of DLA is provided in Algorithm 1. We also plot the difference between Vanilla Linear Attention, Log-Linear Attention, and our Dynamic Linear Attention (DLA) in Figure 2. Specifically, to dynamically determine whether a newly generated token $t$ should be merged into an existing memory state or initiate a new one, we first introduce a new metric named *State Information Score* ($I_t$) to measure the amount of novel information carried by the current token relative to the most recent memory state. Concretely, let $s_t \triangleq \phi(k_t) v_t^\top$ denote the state of new token $t$, and let $S_{t-1}$ denote the previous memory state, which summarizes multiple past tokens. We quantify the information variation between $S_t$ and $S_{t-1}$ as follows:

$$I_t = \frac{\|S_t - S_{t-1}\|_F}{\|S_{t-1}\|_F + \epsilon} \tag{13}$$

In practice, we apply RMSNorm to both $S_t$ and $S_{t-1}$ prior to score computation to further stabilize the scale across layers and timesteps. During inference, we measure the following boundary indicator

$$b_t \triangleq \mathbf{1}[I_t \geq \tau] \tag{14}$$

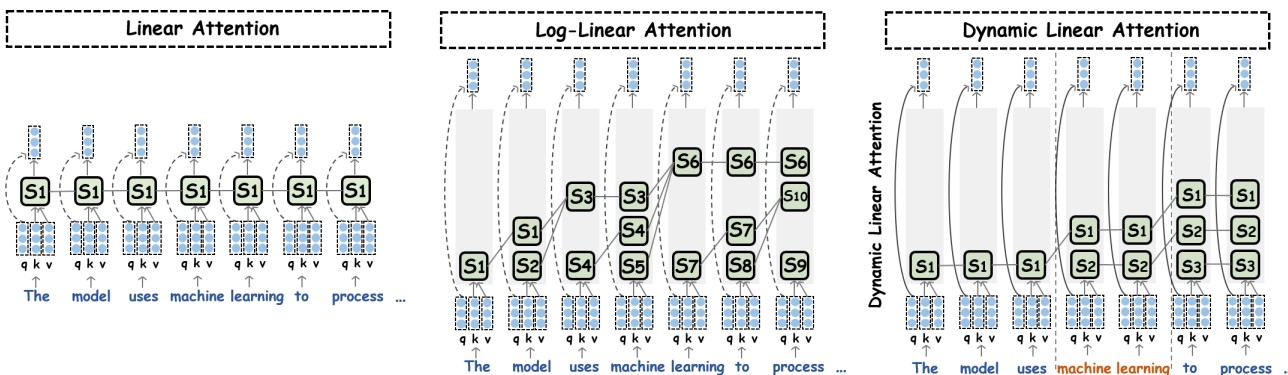

*Figure 2.* Standard linear attention (left) vs. log-linear attention (mid) vs. dynamic linear attention (right). The input consists of query, key, and value vectors.

Let $S_{t-1}^{\text{cur}}$ denote the most recent memory state in the cache. The state update rule at inference is then defined as

$$S_t^{\text{cur}} = \begin{cases} S_{t-1}^{\text{cur}} + s_t, & b_t = 0, \\ S_t, & b_t = 1, \end{cases} \quad (15)$$

where $b_t = 1$ indicates that the current token initiates a new memory state, while $b_t = 0$ continues to accumulate information into the existing state. When $b_t = 1$, the newly created state $S_t$ is appended to the memory cache, preserving the chronological order of states.

We apply soft gating to decide the boundary in a differentiable manner during pre-training and then switch to a hard segmentation strategy during inference to ensure that inference produces a discrete set of memory states aligned with semantic boundaries, while retaining the same information-aware criterion learned during training.

**Discussion.** Theorem 3.1 shows the summarization deviation is dominated by the within-block heterogeneity term in Equation (12). Fixed blocking policies are content-agnostic and therefore may mix tokens from distinct semantic regimes into the same block, which yields a strictly larger deviation bound on non-stationary sequences ( Theorem 3.2). In contrast, DLA monitors token-level representation drift and only merges a new token when the induced increase of heterogeneity is small, using the State Information Score $I_t$ in Equation (13). Therefore, DLA can be viewed as a greedy online strategy that approximately minimizes the dominant term in the deviation bound, while fixed policies ignore it, making it less competitive than DLA.

### 3.2. Capacity-Bounded Memory Modeling

**Motivation:** While the previous design enables flexible and information-aware memory state construction, maintaining an unbounded number of states is impractical for efficient inference, especially in long-context or high-throughput

---

**Algorithm 2** Capacity-Bounded Memory Modeling

1: **Input:** Incoming states $\{S_t\}$, Information Scores $\{\bar{I}_t\}$, Token Counts $\{n_t\}$, Capacity $K$, Queries $\{q_t\}$
2: **Output:** attention outputs $\{o_t\}$
3: $\mathcal{M} \leftarrow [\,]$ {state cache}
4: **for** each $(S_i, \bar{I}_i, n_i)$ in time order **do**
5:    **if** $|\mathcal{M}| = K$ **then**
6:       $(i^\star, i^\star+1) \leftarrow \arg\min_i \frac{\bar{I}_i + \bar{I}_{i+1}}{n_i + n_{i+1}}$
7:       $S_{i^\star} \leftarrow S_{i^\star} + S_{i^\star+1}$
8:       $n_{i^\star} \leftarrow n_{i^\star} + n_{i^\star+1}$
9:       $\bar{I}_{i^\star} \leftarrow \bar{I}_{i^\star} + \bar{I}_{i^\star+1}$
10:      remove $S_{i^\star+1}$ from $\mathcal{M}$
11:    **end if**
12:    append $(S_i, \bar{I}_i, n_i)$ to $\mathcal{M}$
13: **end for**
14: $o_t = \sum_i \phi(q_t)\, S_i$
15: **return** $\{o_t\}$

---

serving scenarios, as dynamic memory growth leads to irregular memory layouts, variable attention costs, and reduced batching efficiency. To address these challenges, it is essential to explicitly limit the number of memory states while preserving the most informative summaries.

**Key Design:** The pseudocode of Capacity-Bounded Memory Modeling of DLA is provided in Algorithm 2. Specifically, DLA maintains a state cache $\mathcal{M} = \{(S_i, n_i, \bar{I}_i)\}_{i=1}^m$ with $m \leq K$, where $S_i \in \mathbb{R}^d$ denotes the $i$-th memory state in chronological order, $n_i$ is the number of tokens summarized by $S_i$, and $\bar{I}_i$ is an aggregated information score of all tokens in this state. We maintain $\bar{I}_i$ as the sum of per-token information scores within each state, such that $\bar{I}_i / n_i$ measures information density. Newly generated tokens are first converted to per-token representations $S_t$, and a tentative state is produced following Section 3.1. The resulting state is appended to the cache, preserving temporal order. When the cache is not full ($m < K$), we simply insert the new

state. When the cache reaches capacity ($m = K$), we trigger a compression step that merges two *adjacent* states to free one slot. Restricting merges to adjacent states preserves the temporal order and avoids distorting positional semantics. Concretely, among all consecutive pairs $(i, i+1)$, we select the pair with the lowest information density:

$$(i^\star, i^\star + 1) = \arg \min_{i \in \{1, \ldots, K-1\}} \frac{\bar{I}_i + \bar{I}_{i+1}}{n_i + n_{i+1}} \quad (16)$$

We then merge them using a summarization operator as in Section 3.1,

$$S_{i^\star} \leftarrow S_{i^\star} + S_{i^\star + 1},$$
$$n_{i^\star} \leftarrow n_{i^\star} + n_{i^\star + 1}, \quad \bar{I}_{i^\star} \leftarrow \bar{I}_{i^\star} + \bar{I}_{i^\star + 1} \quad (17)$$

and shift the remaining states accordingly to keep $m = K - 1$ before inserting the incoming state.

Given the capacity-bounded cache $\mathcal{M}$, we compute the output at time step $t$ by attending over the stored memory states. Let $q_t$ denote the query vector of the current token. The final output is then computed as

$$o_t = \sum_{i=1}^{m} \lambda_{t,i} \, q_t^\top \left( \sum_{s \in \mathcal{C}_i} v_s k_s^\top \right) = \sum_{i=1}^{m} \lambda_{t,i} \, q_t^\top S_i, \quad (18)$$

where $\lambda_{t,i}$ is the weight learned during pre-training with the same shape as the memory capacity. Following (Guo et al., 2025), $\lambda_{t,i}$ is produced by a learned linear layer over the query representation and reused at inference. In this way, DLA provides a unified way to read from a temporally ordered, capacity-bounded memory, enabling stable inference cost while retaining the ability to emphasize informative states during the inference.

# 4. Experiments

## 4.1. Experimental Setups

**Baselines.** We compare DLA against two groups of models: (1) Vanilla linear attention models, including Mamba-2-780M (Dao & Gu, 2024) and Gated DeltaNet-1.3B (Yang et al., 2025). (2) Multi-state linear attention models, including Mamba-2 with Log-linear Attention, and Gated DeltaNet with Log-linear Attention (Guo et al., 2025). Following the design in previous work (Guo et al., 2025), we also compare the DLA version of Mamba-2-780M with full attention Transformers with 24 layers and 778M parameters.

**Datasets.** To demonstrate the generalizability of DLA, we evaluate the performance of DLA on 16 datasets covering three categories, including eight commonsense reasoning datasets (LAMBADA (Paperno et al., 2016), PIQA (Bisk et al., 2020), HellaSwag (Zellers et al., 2019), Wino-Grande (Sakaguchi et al., 2021), OpenBookQA (Mihaylov

et al., 2018), CommonsenseQA (Talmor et al., 2019), ARC-e, and ARC-c (Bhakthavatsalam et al., 2021)), six in-context retrieval datasets (SWDE (Lockard et al., 2019), SQuAD (Rajpurkar et al., 2018), FDA (Arora et al., 2023), TriviaQA (Joshi et al., 2017), Drop (Dua et al., 2019), NQ (Kwiatkowski et al., 2019)), and two long-context datasets (RULER (Hsieh et al., 2024) and LongBench (Bai et al., 2024)). All of the evaluations are conducted using the LM-Evaluation-Harness framework (Gao et al., 2024).

**Implementation Details.** To ensure a fair comparison, we followed the same configuration used in Log-Linear Attention (Guo et al., 2025) to train the full attention Transformer-778M, Mamba-2-780M, Gated DeltaNet-1.3B, and their variants in Log-Linear and DLA forms. Specifically, we perform academic-scale language modeling pretraining from scratch using 50B tokens on the Long-Data-Collections dataset, using a sequence length of 16K. The training loss comparison is provided in Section A.1. We set the capacity of the state cache in DLA to 30, which is the same as the maximum state number in Log-linear attention. All of our experiments are conducted on 4 NVIDIA A100 GPUs.

## 4.2. Overall Comparison

We evaluate the overall performance of DLA from three main aspects: (1) performance on commonsense reasoning tasks, (2) performance on in-context retrieval tasks, and (3) performance on long-context modeling tasks.

**Performance on Commonsense Reasoning.** Following prior work (Dao & Gu, 2024), we evaluate all models on eight commonsense reasoning benchmarks. Results are summarized in Table 1. We make two key observations. First, DLA consistently outperforms both the vanilla and log-linear variants of linear-attention–based models across all tasks. In particular, compared to the log-linear variant, DLA achieves up to 52% and 22% relative accuracy improvement on Mamba-2-780M and Gated DeltaNet-1.3B, respectively. Second, when applied to Mamba-2-780M, DLA also consistently outperforms a full-attention Transformer with a comparable parameter size, demonstrating that DLA can close and even surpass the accuracy gap between linear attention and full attention.

**Performance on In-Context Retrieval Tasks.** Then, we evaluate the models on six in-context retrieval tasks following prior work (Arora et al., 2024). As shown in Table 4, DLA consistently outperforms the baseline methods with at most 49% performance improvement.

**Performance on Long-Context Modeling Tasks.** We next evaluated the models on long-context tasks, including long-context retrieval on RULER with 4k, 8k, 16k length and long-context understanding on LongBench. Due to page limit, the results of 8k and 16k context length on Ruler are

*Table 1.* Performance comparison of `DLA` and baseline methods on zero-shot commonsense reasoning tasks on Mamba-2 (780M) and Gated DeltaNet (1.3B). Commonsense reasoning datasets (LAMBADA, PIQA, HellaSwag, WinoGrande, ARC-e, ARC-c, OpenBookQA, and CommonsenseQA) are measured by accuracy (↑). The best performance is marked in bold. The relative performance gain compared to the best-performing baseline is marked in green inside bracket. The statistic analysis is provided in Section A.6.

| Model | LMB.↑ | PIQA↑ | Hella.↑ | Wino.↑ | ARC-e↑ | ARC-c↑ | OBQA↑ | CSQA↑ | Average↑ |
|---|---|---|---|---|---|---|---|---|---|
| Transformer | 21.8 | 63.1 | 30.3 | 50.9 | 44.2 | 17.7 | 16.8 | 18.0 | 32.9 |
| Mamba-2 | 15.7 | 58.9 | 29.3 | 50.1 | 46.0 | 18.9 | 15.4 | 20.3 | 31.8 |
| w/ Log-linear | 13.2 | 59.7 | 27.8 | 49.5 | 42.3 | 20.1 | 16.0 | 19.1 | 31.0 |
| w/ DLA | **23.9** (↑ 52%) | **63.7** (↑ 7%) | **30.8** (↑ 5%) | **51.5** (↑ 3%) | **48.1** (↑ 5%) | **22.1** (↑ 10%) | **17.4** (↑ 9%) | **21.1** (↑ 4%) | **34.8** (↑ 9%) |
| Gated DeltaNet | 20.3 | 58.8 | 29.6 | 51.3 | 44.7 | 20.2 | 16.0 | 21.3 | 32.8 |
| w/ Log-linear | 19.0 | 60.4 | 28.4 | 51.9 | 44.3 | 20.5 | 15.4 | 21.0 | 32.6 |
| w/ DLA | **24.8** (↑ 22%) | **62.7** (↑ 4%) | **30.3** (↑ 2%) | **52.4** (↑ 2%) | **45.0** (↑ 1%) | **23.0** (↑ 14%) | **17.4** (↑ 9%) | **22.9** (↑ 8%) | **34.8** (↑ 6%) |

*Table 2.* Evaluation results of single-needle tasks (S-NIAH-1–3) and multi-needle tasks (MK-1, MQ, MV) on **RULER** (4K context).

| Model | S-NIAH-1 | S-NIAH-2 | S-NIAH-3 | MK-NIAH-1 | MQ-NIAH | MV-NIAH |
|---|---|---|---|---|---|---|
| Transformer | 91.4 | 71.4 | 73.3 | 62.1 | 54.5 | 36.9 |
| Mamba-2 | 80.5 | 31.0 | 4.7 | 14.8 | 19.3 | 19.4 |
| w/ Log-Linear | 88.7 | 39.4 | 10.6 | 41.6 | 25.2 | 28.3 |
| w/ DLA | 100.0 (↑ 13%) | 69.2 (↑ 76%) | 37.1 (↑ 250%) | 60.3 (↑ 45%) | 42.5 (↑ 67%) | 37.6 (↑ 33%) |
| Gated DeltaNet | 100.0 | 74.0 | 53.2 | 19.1 | 19.5 | 14.8 |
| w/ Log-Linear | 100.0 | 81.6 | 48.8 | 43.8 | 31.3 | 26.6 |
| w/ DLA | 100.0 (↑ 0%) | 98.6 (↑ 21%) | 65.9 (↑ 24%) | 49.3 (↑ 13%) | 34.2 (↑ 9%) | 30.5 (↑ 15%) |

provided in Section A.3. As shown in Table 2 and Table 3, we make two main observations. First, `DLA` substantially improves long-context retrieval performance on RULER across both single-needle and multi-needle settings. Compared to the log-linear variant, `DLA` achieves consistent and often large gains on Mamba-2 and Gated DeltaNet, with particularly pronounced improvements on harder multi-needle tasks (e.g., up to 350% relative improvement on S-NIAH-3 and 67% on MQ-NIAH). These results indicate that `DLA` more effectively preserves and aggregates long-range information under extended contexts. Second, on LongBench, `DLA` consistently outperforms both vanilla and log-linear variants across diverse long-context understanding tasks, including narrative QA, multi-field QA, summarization, and few-shot learning. Notably, `DLA` delivers strong and uniform gains across different task categories, suggesting that the benefits of `DLA` extend beyond retrieval and generalize to complex reasoning and generation under long contexts.

### 4.3. Inference Efficiency of `DLA`

We next evaluate the efficiency of `DLA` from two aspects: (1) efficiency under varying batch sizes and (2) efficiency under varying input context lengths. For each setting, we report both throughput and runtime memory footprint in the prefill and decode stage. Due to page limit, the decode analysis is put in Section A.2.

**Efficiency Under Various Batch Sizes.** Figures Figure 3(a) and (c) report the throughput and runtime memory footprint under varying batch sizes with a fixed context length of 128 and decode length of 1. As batch size increases, both the

log-linear and `DLA` variants exhibit smaller throughput gains and higher memory usage than the vanilla Mamba-2, due to caching multiple summary states. Nevertheless, compared to log-linear attention, `DLA` consistently achieves higher throughput with lower memory consumption, indicating better compute and memory efficiency.

**Efficiency Under Various Context Lengths.** Figures Figure 3(b) and (d) show the throughput and KV memory footprint under varying context lengths with a fixed batch size of 1 and decode length of 1. As context length increases, both log-linear and `DLA` variants incur higher memory usage and limited throughput improvement relative to the vanilla model, again due to maintaining multiple summary states. In contrast, `DLA` consistently outperforms the log-linear variant in throughput while maintaining lower and more stable memory consumption, demonstrating superior efficiency under long-context settings.

### 4.4. Ablation Studies

**Module Sensitivity Study.** We conduct ablation studies to evaluate the separate contribution of the two components of `DLA`. Let `DLA(I)` denote the version of `DLA` with information-aware dynamic state merging only. As shown in Table 5, we have two observations. (1) `DLA(I)` and `DLA` variants consistently outperform Log-Linear variants across all benchmarks. (2) `DLA` consistently outperforms `DLA(I)` across all benchmarks. This result demonstrates the unique contributions of the two components in `DLA`.

**Impact of capacity $k$ and boundary $\tau$.** To study the impact of memory budget $k$ and the merge boundary $\tau$ on performance, we adjust the default budget in `DLA` variant of Mamba-2 from 30 to 20 and 40 and adjust the default boundary from 0.6 to 0.5 and 0.7. We then compare the changes in performance. As shown in Table 6, changes in the memory budget and merge boundary have only a marginal effect on the final performance of `DLA`, indicating that the proposed memory modeling is robust to these two hyperparameters.

*Table 3.* Performance on LongBench datasets (Bai et al., 2024) with different types of tasks.

| | Single-Doc QA | | | Multi-Doc QA | | | Summarization | | | Few-shot Learning | | |
|---|---|---|---|---|---|---|---|---|---|---|---|---|
| **Model** | NQA | QQA | MFQ | HQA | 2WM | Mus | GvR | QMS | MNs | TRC | TQA | SSM |
| Transformer | 8.4 | 9.6 | 19.6 | 11.0 | 20.9 | 6.4 | 12.8 | 9.7 | 9.5 | 21.0 | 43.2 | 14.4 |
| Mamba-2 | 5.1 | 10.6 | 11.9 | 10.2 | 14.5 | 4.7 | 6.3 | 5.5 | 3.3 | 2.3 | 20.9 | 6.8 |
| w/ Log-Linear | 6.8 | 9.6 | 12.2 | 9.5 | 19.0 | 4.4 | 6.4 | 8.1 | 3.5 | 12.4 | 18.6 | 9.8 |
| w/ DLA | **9.4** (↑ 38%) | **11.1** (↑ 5%) | **16.1** (↑ 32%) | **12.5** (↑ 23%) | **23.9** (↑ 26%) | **8.7** (↑ 85%) | **8.3** (↑ 30%) | **12.2** (↑ 51%) | **9.3** (↑ 266%) | **18.7** (↑ 51%) | **31.4** (↑ 50%) | **16.5** (↑ 68%) |
| Gated DeltaNet | 7.2 | 10.3 | 14.0 | 9.8 | 18.5 | 6.3 | 7.2 | 8.4 | 7.6 | 16.5 | 25.3 | 11.0 |
| w/ Log-Linear | 6.9 | 6.1 | 15.4 | 6.1 | 20.2 | 5.5 | 6.1 | 9.9 | 4.3 | 11.0 | 27.7 | 11.2 |
| w/ DLA | **11.3** (↑ 57%) | **15.1** (↑ 47%) | **17.4** (↑ 24%) | **17.2** (↑ 50%) | **25.4** (↑ 26%) | **7.0** (↑ 11%) | **8.7** (↑ 21%) | **11.8** (↑ 19%) | **9.9** (↑ 30%) | **31.2** (↑ 89%) | **41.5** (↑ 50%) | **19.7** (↑ 76%) |

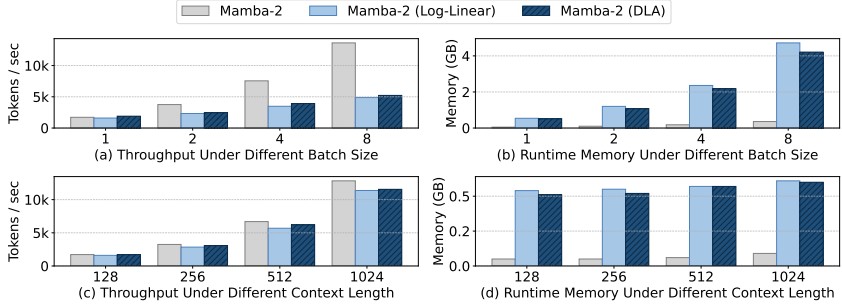

*Figure 3.* Throughput (tokens/sec) and runtime memory consumption (GB) of vanilla, Log-Linear, and `DLA` variants of Mamba-2 (780M) in prefill stage on a single A100 GPU under different bat ch sizes (a, b) and different sequence lengths (c, d).

*Table 4.* Performance on in-context retrieval benchmarks measured by accuracy (↑). The best performance is marked in bold. The relative performance gain compared to the best-performing baseline is marked in green inside bracket.

| Model | SQuAD↑ | TriviaQA↑ | SWDE↑ | FDA↑ | NQ↑ | Drop↑ | Avg.↑ |
|---|---|---|---|---|---|---|---|
| Mamba-2 | 22.4 | 13.6 | 17.4 | 0.0 | 0.2 | 3.4 | 9.5 |
| w/ Log-linear | 7.1 | 9.1 | 15.2 | 0.0 | 0.0 | 6.6 | 6.3 |
| w/ DLA | **28.1** (↑ 25%) | **20.2** (↑ 49%) | **26.3** (↑ 51%) | **0.0** (↑ 0%) | **1.0** (↑ 400%) | **6.6** (↑ 0%) | **13.7** (↑ 44%) |
| Gated DeltaNet | 15.8 | 29.4 | 20.3 | 16.9 | 10.7 | 13.5 | 17.8 |
| w/ Log-linear | 21.7 | 25.3 | 28.8 | 17.7 | 10.1 | 18.3 | 20.3 |
| w/ DLA | **27.3** (↑ 26%) | **33.8** (↑ 15%) | **51.7** (↑ 80%) | **22.2** (↑ 25%) | **12.1** (↑ 13%) | **25.8** (↑ 41%) | **28.8** (↑ 42%) |

*Table 5.* Ablation study of Mamba-2 and Gated DeltaNet with different variants. `DLA(I)` denotes the version of `DLA` with information-aware dynamic state merging only.

| Model | LMB.↑ | PIQA↑ | Hella.↑ | Wino.↑ | RULER↑ |
|---|---|---|---|---|---|
| Mamba-2 | 15.7 | 58.9 | 29.3 | 50.1 | 28.3 |
| w/ Log-linear | 13.2 | 59.7 | 27.8 | 49.5 | 39.0 |
| w/ DLA(I) | 20.1 (↑ 28%) | 61.5 (↑ 3%) | 30.3 (↑ 3%) | 50.9 (↑ 2%) | 44.0 (↑ 13%) |
| w/ DLA | **23.9** (↑ 52%) | **63.7** (↑ 7%) | **30.8** (↑ 5%) | **51.5** (↑ 3%) | **57.8** (↑ 48%) |
| Gated DeltaNet | 20.3 | 58.8 | 29.6 | 51.3 | 46.8 |
| w/ Log-linear | 19.0 | 60.4 | 28.4 | 51.9 | 55.4 |
| w/ DLA(I) | 22.2 (↑ 9%) | 61.3 (↑ 1%) | 29.7 (↑ 5%) | 52.0 (↑ 1%) | 58.4 (↑ 5%) |
| w/ DLA | **24.8** (↑ 22%) | **62.7** (↑ 4%) | **30.3** (↑ 6%) | **52.4** (↑ 1%) | **63.1** (↑ 14%) |

*Table 6.* Ablation study of Mamba-2 `DLA` variant with different memory budget $k$ and merge boundary $\tau$.

| Budget Size | LMB.↑ | PIQA↑ | Hella.↑ | Wino.↑ | RULER↑ |
|---|---|---|---|---|---|
| $k = 20$ | 22.7 | 64.9 | 29.3 | 50.1 | 56.3 |
| $k = 30$ | 23.9 | 63.7 | 30.8 | 51.5 | 57.8 |
| $k = 40$ | 24.2 | 63.3 | 29.8 | 50.9 | 57.7 |
| $\tau = 0.5$ | 20.1 | 63.6 | 29.7 | 51.3 | 57.3 |
| $\tau = 0.6$ | 23.9 | 63.7 | 30.8 | 51.5 | 57.8 |
| $\tau = 0.7$ | 22.5 | 61.7 | 30.1 | 52.3 | 55.9 |

More recent approaches extend linear attention to multistate memory. In particular, Log-Linear Attention (Guo et al., 2025) maintains a logarithmic number of hierarchical states, where tokens are deterministically merged according to a fixed temporal schedule. Despite their effectiveness, these methods rely on *fixed merging policies* that ignore token-level information variation, leaving open the question of how to adaptively control state construction to preserve fine-grained information under long contexts.

## 5. Related Work

To overcome the quadratic bottleneck of softmax attention on long sequences, linear attention and state space models (SSMs) reformulate attention computation to achieve $O(T)$ complexity. Representative methods such as DeltaNet (Yang et al., 2024) and Mamba (Gu & Dao, 2023) compress the entire history into a single recurrent state, continuously merging incoming tokens into a fixed-size summary for inference. To alleviate the resulting over-compression, gating mechanisms (Yang et al., 2025) introduce data-dependent modulation to selectively attenuate obsolete information.

## 6. Conclusion

In this paper, we presented `DLA`, a framework for multistate linear attention. `DLA` replaces fixed merging with information-aware dynamic state construction and uses capacity-bounded memory modeling to keep inference cost predictable. By allocating memory resolution based on token-level information variation, `DLA` improves representation quality while preserving efficiency. We pre-train `DLA` on two linear-attention backbones and evaluate it on 16 datasets across three aspects, where it consistently outperforms state-of-the-art baselines.

## Impact Statement

This paper presents work whose goal is to advance the field of machine learning. There are many potential societal consequences of our work, none of which we feel must be specifically highlighted here.

## Acknowledgement

This work is supported in part by NSF Award NeTS-2312675.

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

# A. Appendix.

### A.1. Training Loss Comparison on Mamba

To better demonstrate the effectiveness of the designs in `DLA`, we also compare the training loss of `DLA`, and log-linear variants on Mamba-2-780M. As shown in Figure 4, we have two main observations. First, `DLA` has the faster convergence speed, which is due to the design of the soft gating mechanism used in the merging of the dynamic state of information. Specifically, during the training stage, we define $g_t = \sigma(I_t)$ and convert the hard segmentation to the following soft gating:

$$S_t = (1 - g_t)(S_{t-1} + s_t) + g_t s_t \tag{19}$$

Although $g_t$ has not been used after training, it forces the model to learn to detect the information of the new token-level state, making the training more smoothly with a better convergence. Second, `DLA` achieves a lower final loss compared with log-linear, which indicates the effectiveness of our proposed components for better modeling the sequence.

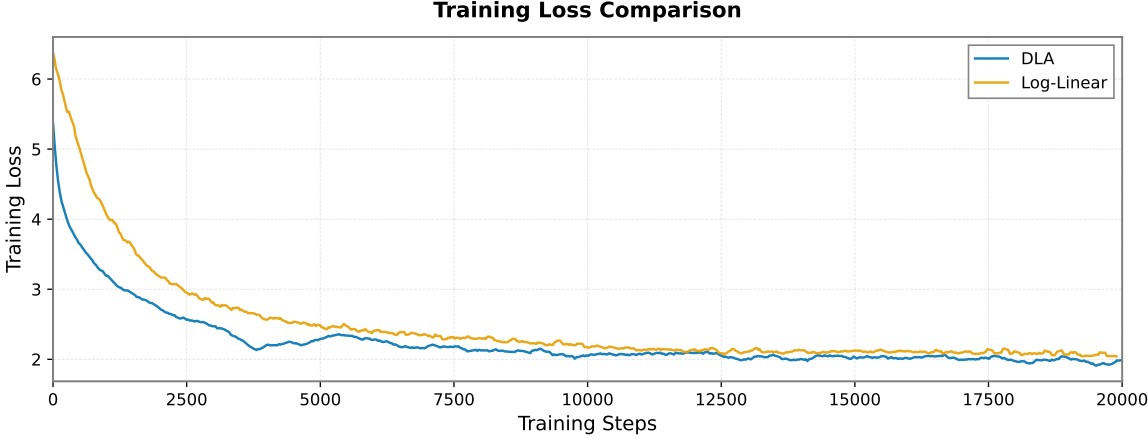

*Figure 4.* Training loss comparison between Log-Linear and `DLA` variants of Mamba-2 (780M).

### A.2. Decoding Throughput Analysis

To further analyze the efficiency of `DLA`, we compare the throughput and run time memory consumption of vanilla, log-linear and `DLA` variants of Mamba-2 (780M) in the decoding stage. We set the input length to 128 and decoding length to 1024. As shown in Figure 5, although not as fast and light-weight as vanilla Mamba-2, `DLA` still achieves a higher decoding throughput with fewer runtime memory consumption compared to log-linear attention.

### A.3. RULER Result on Longer Context

To better demonstrate the generalizability of `DLA`. We report the RULER result on longer context here. The results on 8k and 16k context length are shown in Table 10, similar to the observation in ruler 4k that we obtained in Section 4.2, `DLA` consistently outperforms log-linear across tasks and models.

### A.4. Comparison with Other Efficient Attention Methods

We have also compared `DLA` with other efficient attention methods, including xAttention, MInference, D2O, and FlexPrefill on Transformers-770M. As shown in Table 9. `DLA` still consistently outperforms these new baseline methods.

### A.5. More Ablation Studies

Below we have conducted two additional ablation studies on `DLA`, including merging policy analysis and boundary sensitivity analysis.

**Merging Policy Analysis** We have conducted an ablation comparing several alternative merging policies when the cache reaches capacity, including merging the oldest state, merging the shortest state, and random adjacent merging. The results

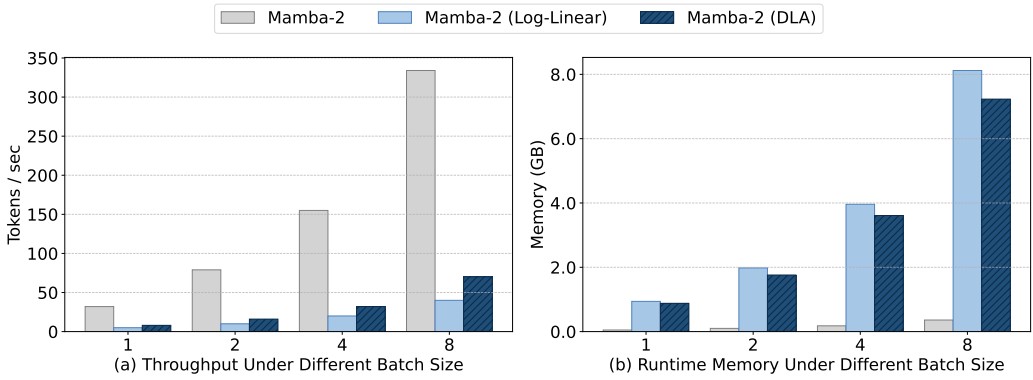

*Figure 5.* Throughput (tokens/sec) and runtime memory consumption (GB) of vanilla, Log-Linear, and `DLA` variants of Mamba-2 (780M) in decoding stage on a single A100 GPU under different batch sizes (a, b).

*Table 7.* Ablation results of different cache eviction strategies.

| Strategy | LMB. | PIQA | Hella. | Wino. | RULER-4K |
|---|---|---|---|---|---|
| Least (LRU) | 23.9 | 63.7 | 30.8 | 51.5 | 57.8 |
| Oldest (FIFO) | 20.8 | 60.4 | 27.6 | 46.2 | 50.5 |
| Shortest | 16.8 | 56.7 | 21.6 | 45.8 | 49.7 |
| Random | 11.6 | 43.7 | 18.2 | 21.9 | 23.4 |

*Table 8.* Ablation results of different cache eviction strategies.

| Strategy | LMB. | PIQA | Hella. | Wino. | RULER-4K |
|---|---|---|---|---|---|
| Least (LRU) | 23.9 | 63.7 | 30.8 | 51.5 | 57.8 |
| Oldest (FIFO) | 20.8 | 60.4 | 27.6 | 46.2 | 50.5 |
| Shortest | 16.8 | 56.7 | 21.6 | 45.8 | 49.7 |
| Random | 11.6 | 43.7 | 18.2 | 21.9 | 23.4 |

are provided in Table 8. Across these variants, we consistently observe that merging the least informative adjacent states (the default policy of `DLA`) achieves the best performance across datasets.

**Boundary Sensitivity Analysis** We have analyzed the segmentation behavior of `DLA` on long sequences (RULER-4K) and reported average number of states per 1K tokens, and how often the cache hits capacity K as shown in Table 8. We obtained the following observations. First, the average number of maintained states per 1K tokens is stable, indicating that DLA consistently operates under the intended budget rather than producing erratic segmentations. Second, the cache hits the capacity K at a controlled frequency, confirming that the capacity-bounded design is actively utilized while avoiding excessive merging. Last, the variance of state counts across sequences is low, suggesting that segmentation is not highly sensitive to specific inputs or noise.

## A.6. Statistical Robustness

For each method, we repeat evaluation with $n$ different random seeds while keeping all settings fixed. As shown in Table 11 and Table 12, we report mean $\mu$ and standard deviation $\sigma$ across runs, and additionally provide 95% confidence intervals (CI) using $\mu \pm t_{0.975,\,n-1} \cdot \sigma/\sqrt{n}$.

*Table 9.* Evaluation results on commonsense reasoning and RULER-4K benchmarks.

| Method | LMB. | PIQA | Hella. | Wino. | RULER-4K | ARC-e | ARC-c | OBQA | CSQA |
|---|---|---|---|---|---|---|---|---|---|
| Baseline | 21.8 | 63.1 | 30.3 | 50.9 | 31.4 | 44.2 | 17.7 | 16.8 | 18.0 |
| XAttention | 21.7 | 62.8 | 30.1 | 49.6 | 32.8 | 42.6 | 16.2 | 16.6 | 20.2 |
| MInference | 22.5 | 62.1 | 30.5 | 51.1 | 42.0 | 43.0 | 18.1 | 15.2 | 19.1 |
| D2O | 21.4 | 58.7 | 30.7 | 49.3 | 15.1 | 39.9 | 10.5 | 15.6 | 19.5 |
| FlexPrefill | 20.0 | 63.1 | 30.4 | 51.2 | 43.7 | 41.0 | 21.6 | 17.0 | 19.5 |
| DLA | **23.9** | **63.7** | **30.8** | **51.5** | **57.8** | **48.1** | **22.1** | **17.4** | **21.1** |

*Table 10.* Results on longer context.

| Model | S-NIAH-1 (pass-key retrieval) | | | S-NIAH-2 (number in haystack) | | | S-NIAH-3 (uuid in haystack) | | |
|---|---|---|---|---|---|---|---|---|---|
| | 4K | 8K | 16K | 4K | 8K | 16K | 4K | 8K | 16K |
| Transformer | 91.4 | 73.4 | 69.2 | 71.4 | 68.0 | 65.4 | 73.3 | 64.1 | 42.0 |
| Mamba-2 | 80.5 | 53.1 | 16.5 | 31.0 | 15.4 | 3.2 | 4.7 | 1.9 | 0.7 |
| w/ Log-Linear | 88.7 | 64.8 | 34.9 | 39.4 | 34.1 | 10.5 | 10.6 | 5.6 | 1.8 |
| w/ DLA | 100.0 | 99.2 | 56.8 | 69.2 | 71.0 | 21.3 | 37.1 | 20.7 | 8.9 |
| Gated DeltaNet | 100.0 | 100.0 | 100.0 | 74.0 | 47.6 | 4.5 | 33.2 | 17.3 | 1.8 |
| w/ Log-Linear | 100.0 | 100.0 | 100.0 | 81.6 | 56.2 | 8.7 | 48.8 | 21.4 | 4.6 |
| w/ DLA | 100.0 | 100.0 | 100.0 | 98.6 | 83.5 | 25.4 | 49.1 | 21.8 | 13.4 |

| Model | MK-NIAH-1 (multi-key line retrieval) | | | MQ-NIAH (multi-query) | | | MV-NIAH (multi-value) | | |
|---|---|---|---|---|---|---|---|---|---|
| | 4K | 8K | 16K | 4K | 8K | 16K | 4K | 8K | 16K |
| Transformer | 62.1 | 54.7 | 45.8 | 54.5 | 46.6 | 27.4 | 36.9 | 29.3 | 21.9 |
| Mamba-2 | 14.8 | 9.7 | 5.4 | 19.3 | 12.9 | 1.6 | 19.4 | 10.8 | 5.3 |
| w/ Log-Linear | 41.6 | 24.6 | 14.7 | 25.2 | 19.6 | 5.8 | 28.3 | 16.7 | 8.4 |
| w/ DLA | 60.3 | 34.1 | 24.2 | 42.5 | 27.8 | 7.6 | 37.6 | 31.5 | 23.8 |
| Gated DeltaNet | 19.1 | 15.7 | 5.6 | 19.5 | 14.4 | 8.7 | 14.8 | 10.9 | 6.4 |
| w/ Log-Linear | 43.8 | 17.1 | 7.9 | 31.3 | 21.4 | 9.6 | 26.6 | 21.8 | 14.7 |
| w/ DLA | 49.3 | 27.9 | 11.1 | 34.2 | 24.7 | 10.4 | 30.5 | 28.6 | 17.7 |

*Table 11.* Statistical robustness for Table 1. We report mean $\pm$ std over $n = 3$ independent runs with different random seeds.

| Model | LMB. | PIQA | Hella. | Wino. | ARC-e | ARC-c | OBQA | CSQA | Average |
|---|---|---|---|---|---|---|---|---|---|
| Transformer | $21.8_{\pm 0.4}$ | $63.1_{\pm 0.3}$ | $30.3_{\pm 0.5}$ | $50.9_{\pm 0.4}$ | $44.2_{\pm 0.5}$ | $17.7_{\pm 0.3}$ | $16.8_{\pm 0.4}$ | $18.0_{\pm 0.4}$ | $32.9_{\pm 0.3}$ |
| Mamba-2 | $15.7_{\pm 0.5}$ | $58.9_{\pm 0.4}$ | $29.3_{\pm 0.4}$ | $50.1_{\pm 0.3}$ | $46.0_{\pm 0.5}$ | $18.9_{\pm 0.4}$ | $15.4_{\pm 0.3}$ | $20.3_{\pm 0.4}$ | $31.8_{\pm 0.3}$ |
| w/ Log-linear | $13.2_{\pm 0.4}$ | $59.7_{\pm 0.3}$ | $27.8_{\pm 0.5}$ | $49.5_{\pm 0.4}$ | $42.3_{\pm 0.6}$ | $20.1_{\pm 0.4}$ | $16.0_{\pm 0.3}$ | $19.1_{\pm 0.4}$ | $31.0_{\pm 0.3}$ |
| w/ DLA | $23.9_{\pm 0.4}$ | $63.7_{\pm 0.3}$ | $30.8_{\pm 0.4}$ | $51.5_{\pm 0.3}$ | $48.1_{\pm 0.4}$ | $22.1_{\pm 0.3}$ | $17.4_{\pm 0.3}$ | $21.1_{\pm 0.4}$ | $34.8_{\pm 0.2}$ |
| Gated DeltaNet | $20.3_{\pm 0.4}$ | $58.8_{\pm 0.4}$ | $29.6_{\pm 0.4}$ | $51.3_{\pm 0.3}$ | $44.7_{\pm 0.5}$ | $20.2_{\pm 0.4}$ | $16.0_{\pm 0.3}$ | $21.3_{\pm 0.4}$ | $32.8_{\pm 0.3}$ |
| w/ Log-linear | $19.0_{\pm 0.4}$ | $60.4_{\pm 0.3}$ | $28.4_{\pm 0.4}$ | $51.9_{\pm 0.3}$ | $44.3_{\pm 0.4}$ | $20.5_{\pm 0.3}$ | $15.4_{\pm 0.3}$ | $21.0_{\pm 0.4}$ | $32.6_{\pm 0.3}$ |
| w/ DLA | $24.8_{\pm 0.3}$ | $62.7_{\pm 0.3}$ | $30.3_{\pm 0.4}$ | $52.4_{\pm 0.3}$ | $45.0_{\pm 0.4}$ | $23.0_{\pm 0.3}$ | $17.4_{\pm 0.3}$ | $22.9_{\pm 0.3}$ | $34.8_{\pm 0.2}$ |

*Table 12.* 95% confidence interval (CI) half-widths corresponding to Table 11.

| Model | LMB. | PIQA | Hella. | Wino. | ARC-e | ARC-c | OBQA | CSQA | Average |
|---|---|---|---|---|---|---|---|---|---|
| Transformer | 1.0 | 0.7 | 1.2 | 1.0 | 1.2 | 0.7 | 1.0 | 1.0 | 0.7 |
| Mamba-2 | 1.2 | 1.0 | 1.0 | 0.7 | 1.2 | 1.0 | 0.7 | 1.0 | 0.7 |
| w/ Log-linear | 1.0 | 0.7 | 1.2 | 1.0 | 1.5 | 1.0 | 0.7 | 1.0 | 0.7 |
| w/ DLA | 1.0 | 0.7 | 1.0 | 0.7 | 1.0 | 0.7 | 0.7 | 1.0 | 0.5 |
| Gated DeltaNet | 1.0 | 1.0 | 1.0 | 0.7 | 1.2 | 1.0 | 0.7 | 1.0 | 0.7 |
| w/ Log-linear | 1.0 | 0.7 | 1.0 | 0.7 | 1.0 | 0.7 | 0.7 | 1.0 | 0.7 |
| w/ DLA | 0.7 | 0.7 | 1.0 | 0.7 | 1.0 | 0.7 | 0.7 | 0.7 | 0.5 |

