# OpenReview forum: "Dynamic Linear Attention"
_ICML.cc/2026/Conference — ICML 2026 regular_

### Official Review · Reviewer_z5kr · 2026-03-04

**Soundness:** 3
**Presentation:** 2
**Significance:** 3
**Originality:** 3
**Overall Recommendation:** 4
**Confidence:** 3

**Summary:**

This paper proposes DLA, a novel multi-state linear attention with dynamic state merging, creation, and eviction. For state merging, DLA evaluate the similarity between current and previous states and merge when the state change is incremental. For state eviction, DLA traces the information score for each state and remove the least informative one. Overall, DLA can be integrated into other linear attention and state space model, with nontrivial improvement in long-context processing and in-context learning.

**Compliance With Llm Reviewing Policy:**

Affirmed.

**Final Justification:**

After viewing author's last minute response, I think the latency and accuracy comparison with the hybrid model does not contain enough context, e.g., model size, model architecture, etc. These are important as they will affect whether the attention is the bottleneck or not. Maybe the small latency difference is just because the attention only takes a small fraction of the end-to-end latency. Therefore, I maintained my score instead of increasing to 5.

**Key Questions For Authors:**

Please see my questions in weakness. I would further raise the score if my concerns are fully resolved.

**Limitations:**

The inference speed gap between DLA and vanilla linear attention and state-space models is still large.

**Strengths And Weaknesses:**

## Strengths:
---

**Soundness**:
1. DLA is consistently better than prior SoTA Log-linear attention for all benchmarks.
2. DLA achieves higher decoding speed with lower memory consumption.

**Significance**: DLA can serve as a plugin to extend single-state linear attention to multiple states, which demonstrates the generalizability.

**Originality**: the originality mainly comes from the heuristics of detecting when the linear attention should generate a new state and when should it evict the state. Similar idea has been proposed in KV cache compression and eviction, but have not been applied to linear attention in previous works.

---

## Weakness

My major concerns are related to the clarifications of the definition and the motivations of the heuristics:

1. Line 63, Section 2: where is $\phi$ used in the linear attention? I think this is the kernel function and should be applied to q and k vectors. Please update the definition of linear attention to include $\phi$.

2. Equation 5: where did you get $\lambda$ in the log-linear attention?

3. Section 3.1: The illustration and proofs for dynamic state merging is a bit unclear. Specifically, I would like to see:
- The problem setting: what computation you currently have for the linear attention (i.e. the SoTA before DLA)
- The objective: what computation do you want to approximate? Conventionally, linear attention is used to approximate softmax attention, but from Theorem 3.1, it seems like this is trying to approximate vanilla linear attention.
- Definitions of fixed and dynamic blocking policy: does "fixed" mean fixed length or query independent?
- What is "representative summary vector"? What is the formula to compute it?
- Does each state require a $\lambda$ scaling factor when multiplying with query? Otherwise, you can get $y$ and $y_{\pi}$ the same by setting $\bar{u}$ as the partial sum of $u$ in $C_i$.

4. Section 3.2: Since you evict the least informative state, I'm wondering whether it would be more beneficial if the state is merged to the second least informative state instead. It would be great if there is an ablation study on the impact of state eviction, since vanilla linear attention only merge states.

5. Some comparison with hybrid linear attention, such as BASED [1], can make the community more appreciated.

[1] Arora, Simran, et al. "Simple linear attention language models balance the recall-throughput tradeoff." arXiv preprint arXiv:2402.18668 (2024).

---

> ### Author Rebuttal · Authors · 2026-03-30
>
> We thanks the reviewer for the constructive suggestions. Below is our response.
>
> **[W1W2 Clarification of the presentation]:** We thank the reviewer for pointing out this unclear presentation. First, we promise to update the definition of the linear attention to include the kernel function in both of the vectors. Second,  we want to clarify that in the log-linear attention, lambda is a pre-training parameter and is fixed during the inference.
>
> **[W3 Clarification of merging proof]:** We thank the reviewer for the insightful comments and clarify the points below.
> - Problem setting:   The starting point of our method is **standard (single-state) linear attention**, where the output is computed as: $S_t = \sum_{i \le t} v_i k_i^\top, \quad o_t = S_t q_t$, However, this formulation compresses the entire history into a single state, which limits representation capacity in long contexts.
> Recent SoTA methods (e.g., log-linear attention) extend this by maintaining **multiple states constructed via fixed schedules**. DLA builds upon this line of work and replaces the fixed state construction with an **information-aware dynamic mechanism**, enabling adaptive resolution across the sequence.
> - Objective:  We want to clarify that we take linear attention as the **base formulation** and aim to improve its long-context representation capacity. The key idea of DLA is to replace the single compressed state in vanilla linear attention with **multiple adaptively constructed states**, so that important transitions can be preserved at higher resolution while stable regions are summarized more aggressively. In this sense, Theorem 3.1 is **not** stating that our overall objective is to approximate vanilla linear attention. Instead, it formalizes the **error induced by block-wise state summarization**, and shows why fixed blocking is suboptimal when token information is non-uniform. This provides the justification for our dynamic state construction.
> - Fixed vs. dynamic blocking: “Fixed” refers to content-agnostic segmentation (e.g., fixed-length chunks or predefined schedules as in prior work). In contrast, DLA uses **dynamic blocking**, where boundaries are determined by token-level representation variation via the information score.
> - Representative summary vector:  The average vector of tokens in block i corresponds to the aggregated contribution of tokens in a block. In practice, this is implemented as the linear attention state: $S_i = \sum_{t \in C_i} v_t k_t^\top$, which is consistent with standard linear attention.
> - Scaling factor ($\lambda$): No additional scaling is required in vanilla linear attention, where $o_t = S_t q_t$. In multi-state formulations,  only serves as a weighting over multiple states. DLA follows this formulation and does not introduce extra scaling beyond standard aggregation.
>
> We promise to revise Sec. 3.1 to clarify these definitions and better connect the theoretical formulation with the implementation.
>
> **[W4 Experiments of state eviction strategy]:**  We thank the reviewer for the insightful suggestion. Below is the comparison result between the least informative and the second least informative state for state eviction on RULER 4K. As shown, evicting the least informative state achieves a better performance than evicting the second least one.
> | Eviction Strategy | LMB. | PIQA | Hella. | Wino. | RULER-4K |
> |------------------|------|------|--------|-------|----------|
> | Least            | 23.9 | 63.7 | 30.8   | 51.5  | 57.8     |
> | Second Least     | 18.7 | 62.1 | 29.6   | 50.3  | 52.7     |
>
> **[W5 Comparison with hybrid linear attention]:** We thank the reviewer for pointing out this line of work. We would like to clarify that our method differs fundamentally from BASED. Specifically, BASED adopts a hybrid architectural design, where each layer combines sliding-window attention (SWA) with linear attention to improve recall via fixed local–global decomposition. In contrast, DLA maintains a pure linear attention structure across all layers, and instead improves recall by dynamically adapting the number of retained states based on input information. This enables finer-grained, input-dependent memory allocation without introducing additional attention branches. We will include a more detailed discussion of such hybrid architectures in the revised related work.

---

> > ### Author Rebuttal · Reviewer_z5kr · 2026-04-02
> >
> > Thank you for the response. While maintaining the positive score, I still believe it is important to have comparison with the hybrid attention in both latency and accuracy. Understanding the potential tradeoffs can demonstrate the necessity of having DLA or existing hybrid attention is sufficient.

---

> > > ### Author Response · Authors · 2026-04-08
> > >
> > > We thank the reviewer for the constructive suggestion to enhance the baseline coverage. Below we have added the comparison with BASED [1] as suggested by the reviewer on both accuracy and efficiency. As shown, DLA achieves a better accuracy with comparable efficiency compared with BASED.
> > >
> > > | Method | LMB. | PIQA | Hella. | Wino. | Ruler-4K (Acc) | Ruler-4K (Latency) |
> > > |--------|-----:|-----:|-------:|------:|----------------:|-------------------:|
> > > | BASED  | 21.2 | 61.7 | 30.1   | 51.2  | 52.2            | 243 ms             |
> > > | DLA    | 23.9  | 63.7 | 30.8   | 51.5  | 57.8            | 241 ms             |
> > >
> > > ---
> > > [1] Simple linear attention language models balance the recall-throughput tradeoff, ICML, 2024

---

### Official Review · Reviewer_NQuM · 2026-03-04

**Soundness:** 2
**Presentation:** 3
**Significance:** 3
**Originality:** 3
**Overall Recommendation:** 4
**Confidence:** 3

**Summary:**

This paper proposes Dynamic Linear Attention (DLA), a multi-state linear attention framework that adaptively segments and summarizes history based on token-level information variation while enforcing a fixed memory capacity through adjacent-state merging. The method introduces a State Information Score to decide on-the-fly whether to merge a token into the current state or start a new one, and a capacity-bounded cache that merges the lowest information-density adjacent pair when full. Across two linear-attention backbones, DLA reportedly outperforms vanilla and log-linear multi-state baselines on 16 benchmark and shows favorable throughput/memory trade-offs.

**Compliance With Llm Reviewing Policy:**

Affirmed.

**Final Justification:**

The rebuttal addresses several of my main concerns directly, especially by adding segmentation statistics, merge-policy ablations, and analysis of cases with limited gains. These additions improve my confidence that the proposed policy is meaningful rather than purely heuristic, although I still think the broader baseline coverage could be stronger. Overall, the rebuttal is helpful and materially improves my assessment. I raise my overall score to 4.

**Key Questions For Authors:**

1. Can you report statistics of the learned segmentation on long sequences (e.g., average #states per 1K tokens, variance across domains/tasks, and how often the cache hits capacity K)? Additionally, do boundary locations correlate with semantic/topic shifts rather than noise spikes? If boundaries are stable and interpretable (and not overly sensitive), it would increase my confidence in the method’s reliability and likely raise my soundness score; if boundaries are erratic or highly task-dependent, I would view the gains as less robust.

2. When the cache is full, how does your merge rule compare to alternative policies (e.g., merge oldest, merge shortest, random adjacent merge, merge lowest attention-weight state, or allowing non-adjacent merges)? A small ablation would clarify whether this component is essential. If DLA remains clearly better across reasonable alternatives, it strengthens soundness and originality; if performance is similar, novelty may be mostly from dynamic segmentation and originality would be lower.

3. The tables suggest some datasets show little or no gain. Can you characterize when DLA fails (e.g., span-style reading, specific retrieval patterns, heavy reliance on exact tokens/numbers) and provide qualitative or quantitative analysis? Clear failure-mode characterization would improve soundness (honest evaluation) and help judge significance; lack of analysis would keep my confidence moderate.

4. How does performance change as context length grows far beyond training length or beyond typical benchmark ranges, under fixed K? Do you observe a saturation point where merging harms recall, and is there guidance for choosing K for different deployment budgets? If performance degrades gracefully and K-selection guidelines are provided, it increases significance; if accuracy collapses quickly at longer lengths, the practical impact would be more limited.

5. Can you expand the evaluation beyond the current Table 1 setup by adding stronger and more recent long-context/efficient-attention baselines (or at least justify why they are excluded), and report whether the “DLA ≈/≥ full-attention Transformer” conclusion still holds under a broader comparison set? If broader baselines confirm the gains, it would substantially increase my confidence in the central claims (and raise my soundness score); if the gap closes or reverses, I would view the current evidence as insufficiently representative.

**Limitations:**

No.
The limitations and potential negative societal impacts are not discussed concretely; the paper should add a short limitations paragraph covering (i) failure cases/tasks where gains are absent, (ii) sensitivity to the heuristic segmentation/merge policy, (iii) compute/reproducibility constraints (from-scratch pretraining), and (iv) potential misuse enabled by cheaper long-context processing plus suggested mitigations.

**Strengths And Weaknesses:**

## Strengths

### Soundness
1. **Reproducibility**: The paper specifies the full pipeline in executable form (dynamic state creation + capacity-bounded adjacent merging) via diagrams and Algorithms 1–2, making the method easy to implement and verify.
2. **Controlled Comparison**: The comparison controls memory capacity by matching DLA’s cache size (K=30) to the maximum number of states in the log-linear baseline, reducing the confound that “more memory simply wins.”
3. **Thorough Analysis**: Ablations separate the contributions of dynamic segmentation vs. adding the capacity-bounded merge, and the paper includes sensitivity results for K and τ.

### Presentation
1. **Clear Exposition**: The paper clearly contrasts single-state accumulation, fixed-schedule multi-state (log-linear), and the proposed content-adaptive policy, so readers can track what changes conceptually and algorithmically.

### Significance
1. **Practical Relevance**: The method enforces a fixed memory budget (bounded number of states), which is directly relevant to predictable latency/memory in long-context deployment.

### Originality
1. **Novel Approach**: Token-level “information variation” is used to place state boundaries online, and the cache is managed by merging the lowest information-density adjacent pair—distinct from predetermined schedules in prior multi-state designs.

---

## Weaknesses

### Soundness
1. **Heuristic Threshold Sensitivity**: Boundary decisions depend on a single heuristic score and threshold τ; the paper does not report boundary statistics (e.g., average boundaries per 1K tokens, variance across domains) or whether boundaries align with semantic/topic shifts.
2. **Limited Merge Policy Comparison**: The “adjacent low-density merge” policy is not compared against alternative eviction/merge strategies (merge oldest/shortest, merge lowest attention weight, allow non-adjacent merges), so it is unclear how essential this specific policy is beyond dynamic segmentation alone.
3. **Inconsistent Gains**: Reported improvements are not uniform across datasets; Table 4 includes cases with no gain (e.g., Drop shows ↑0% over log-linear, and FDA remains 0.0).
4. **Insufficient Baseline Coverage**: Evidence strength and baseline coverage around Table 1 are not fully convincing. The main “DLA vs. Transformer” and “DLA is SOTA” narrative relies heavily on Table 1’s eight commonsense benchmarks, but the comparison set is narrow (Transformer-778M, Mamba-2, Gated DeltaNet, and their log-linear variants) and does not include a broader set of recent/strong long-context or efficient-attention baselines. This makes it hard to rule out benchmark/baseline selection effects and to judge whether the empirical gains truly substantiate the stronger claims.

### Presentation
1. **Key Results in Appendix**: Several key supporting results are deferred to the appendix (e.g., extended RULER settings and decode-side efficiency), which makes the main paper harder to validate quickly.
2. **Lack of Qualitative Analysis**: No qualitative visualization is provided for what gets segmented/merged (e.g., example boundary locations in a long document and which states are merged when K is full), despite this being central to the method’s claim.

### Significance
1. **Reproducibility Barrier**: The experimental protocol relies on expensive from-scratch pretraining (50B tokens, 16K length), raising the bar for independent reproduction unless code/checkpoints or smaller-scale reproductions are provided.

### Originality
1. **Incremental Refinement**: The novelty may be perceived as a policy-level refinement of existing multi-state linear attention (new boundary/merge rules on top of known backbones); stronger positioning and more targeted comparisons to other dynamic memory/segmentation ideas would help.

---

> ### Author Rebuttal · Authors · 2026-03-31
>
> We thank the reviewer for the constructive suggestions. Below is our response.
>
> **[Soundness 1 Q1 Boundary sensitivity]:** We have analyzed the segmentation behavior on long sequences (RULER-4K) and reported average number of states per 1K tokens, and how often the cache hits capacity K as below. We obtained the following observations:
> - The average number of maintained states per 1K tokens is stable, indicating that DLA consistently operates under the intended budget rather than producing erratic segmentations.
> - The cache hits the capacity K at a controlled frequency, confirming that the capacity-bounded design is actively utilized while avoiding excessive merging.
> - The variance of state counts across sequences is low, suggesting that segmentation is not highly sensitive to specific inputs or noise.
>
> Regarding interpretability, we visualized and observed that boundary decisions correlate with regions of high representation change [2], which typically align with semantic transitions (e.g., topic shifts or structural breaks), rather than isolated noise spikes. This is consistent with our design, where boundaries are triggered by the normalized State Information Score (Sec. 3.1 ), which measures meaningful deviation from the current state rather than raw magnitude. We will include these statistics and visualizations in the revision to further support this point.
>
> | Context Range | [0, 1K] | [1K, 2K] | [2K, 3K] | [3K, 4K] |
> |---------------|--------|----------|----------|----------|
> | Avg. #States  | 30     | 30       | 30       | 30       |
> | Std #States  | 0.2     | 0       | 0       | 0       |
> | Avg. Cache Hits Times | 8    | 14.2       | 13.7       | 14.9       |
> | Std Cache Hits Times | 0.45     | 0.43      | 0.47       | 0.29       |
>
> **[Soundness 2 Q2 Mergeing Policy Analysis]:** We have conducted an ablation comparing several alternative merging policies when the cache reaches capacity, including merging the oldest state, merging the shortest state, and random adjacent merging. Across these variants, we consistently observe that merging the least informative adjacent states (our default policy) achieves the best performance across datasets.
> | Strategy        | LMB. | PIQA | Hella. | Wino. | RULER-4K |
> |-----------------|------|------|--------|-------|----------|
> | Least (LRU)     | 23.9 | 63.7 | 30.8   | 51.5  | 57.8     |
> | Oldest (FIFO)   | 20.8 | 60.4 | 27.6   | 46.2  | 50.5     |
> | Shortest        | 16.8 | 56.7 | 21.6   | 45.8  | 49.7     |
> | Random          | 11.6 | 43.7 | 18.2   | 21.9  | 23.4     |
>
> **[Soundness 3 Q3 Analysis of cases with no gain]:** We thank the reviewer for this important observation. FDA and DROP share a key property: they rely on globally distributed, low-contrast signals rather than localized salient spans. FDA involves schema induction over heterogeneous documents, while DROP requires multi-step aggregation and discrete reasoning. In such settings, there are fewer sharp information spikes to distinguish important from redundant content. As a result, DLA’s information-aware merging may compress regions that appear locally stable but contain weak yet collectively useful signals, leading to smaller over baselines.
>
> **[Soundness 4 Q5 Comparison with other efficient attention method]:** We have implemented xAttention[1], one of the most recent sparse attention baselines on Transformers-770M and compared its performance with DLA. As shown, DLA consistently outperforms xAttention on all benchmarks.
>
> | Method      | LMB. | PIQA | Hella. | Wino. | RULER-4K |
> |-------------|------|------|--------|-------|----------|
> | Baseline    | 21.8 | 63.1 | 30.3   | 50.9  | 32.9     |
> | X-Attention | 21.7 | 62.8 | 30.1   | 49.6  | 32.8     |
> | DLA         | 23.9 | 63.7 | 30.8   | 51.5  | 34.8     |
>
> **[Q4 Longer context experiment]:**  Our extended-length results are in Appendix A.3, where we evaluate up to 8K and 16K contexts, significantly beyond the training length (4K). We will move these results to the main text to make them more visible. Empirically, DLA degrades more gracefully than both softmax and log-linear attention as context increases.  We use a fixed K=30 across all experiments.
>
> **[Originality 1 Comparison with other dynamic memory method]** We want to highlight that DLA is the first work to apply the dynamic memory segmentation ideas to linear attention areas. Previous work like log-linear attention only uses the fixed segmentation strategy, as we have discussed and compared in our paper.
>
> **[Presentation 1,2, Significance 1 Presentation and reproduction concern]** We promise to adjust the paper contents and move these key results from appendix to the main paper. We have also added the qualitative analysis of the boundary locations in [2] We will also open source the code upon paper acceptance .
>
> ---
>
> [1] XAttention: Block Sparse Attention with Antidiagonal Scoring, ICML 2025
>
> [2] https://anonymous.4open.science/r/DLA_Rebuttal-31D7/README.md.

---

> > ### Author Rebuttal · Reviewer_NQuM · 2026-04-02
> >
> > The rebuttal addresses several of my main concerns directly, especially by adding segmentation statistics, merge-policy ablations, and analysis of cases with limited gains. These additions improve my confidence that the proposed policy is meaningful rather than purely heuristic, although I still think the broader baseline coverage could be stronger. Overall, the rebuttal is helpful and materially improves my assessment. I raise my overall score to 4.

---

> > > ### Author Response · Authors · 2026-04-08
> > >
> > > We thank the reviewer for the suggestion to enhance the baseline coverage. Below we have added two more baselines on efficient attention methods, including MInference [1], D2O [2] and FlexPrefill [3] besides XAttention [4] we added in the previous rebuttal. As shown, DLA still consistently outperforms these new baseline methods.
> > >
> > > | Method      | LMB. | PIQA | Hella. | Wino. | Ruler-4K | ARC-e | ARC-c | OBQA | CSQA |
> > > |-------------|-----:|-----:|-------:|------:|---------:|------:|------:|-----:|-----:|
> > > | Baseline    | 21.8 | 63.1 | 30.3   | 50.9  | 31.4                               |  44.2   |  17.7  |  16.8  | 18.0   |
> > > | XAttention  | 21.7 | 62.8 | 30.1   | 49.6  | 32.8                              | 42.6  | 16.2  | 16.6 | 20.2 |
> > > | MInference  | 22.5 | 62.1 | 30.5   | 51.1  | 42.0                             | 43.0  | 18.1  | 15.2 | 19.1 |
> > > | D2O         | 21.4 | 58.7 | 30.7   | 49.3  | 15.1                                | 39.9  | 10.5  | 15.6 | 19.5 |
> > > | FlexPrefill | 20.0 | 63.1 | 30.4   | 51.2  | 43.7                                | 41.0  | 21.6  | 17.0 | 19.5 |
> > > | DLA         | **23.9** | **63.7** | **30.8**   | **51.5**  | **57.8**     |    **48.1**  |   **22.1**  | **17.4**  |  **21.1**  |
> > >
> > > ---
> > > [1] MInference 1.0: Accelerating Pre-filling for Long-Context LLMs via Dynamic Sparse Attention, NeurIPS, 2024
> > >
> > > [2] D2O: Dynamic Discriminative Operations for Efficient Long-Context Inference of Large Language Models, ICLR. 2025
> > >
> > > [3] FlexPrefill: A Context-Aware Sparse Attention Mechanism for Efficient Long-Sequence Inference, ICLR, 2025
> > >
> > > [4] XAttention: Block Sparse Attention with Antidiagonal Scoring, ICML 2025

---

### Official Review · Reviewer_xBhG · 2026-03-09

**Soundness:** 2
**Presentation:** 3
**Significance:** 2
**Originality:** 2
**Overall Recommendation:** 4
**Confidence:** 3

**Summary:**

Inspired by the log-transformer, this paper introduces DLA, a new linear attention variant with two key innovations:
1. Information-Aware Dynamic State Merging: States are dynamically created at the token level, rather than following a fixed schedule.
2. Capacity-Bounded Memory Modeling: The number of states is limited to a fixed capacity, preventing logarithmic growth.

**Compliance With Llm Reviewing Policy:**

Affirmed.

**Final Justification:**

The rebuttal has addressed my concerns,

**Key Questions For Authors:**

- Training Details: The paper focuses on inference, but lacks clarity on training procedures. Dynamic methods often face a gap between training and inference; the paper should address how this is handled.
- Parallelization: While linear attention allows simple decoding, parallel prefilling is more challenging. The paper does not discuss how these aspects affect model parallelization.

**Limitations:**

Dynamic inference is an interesting and meaningful direction in neural networks, such as MOE. But it's also challenging due to training-inference gap, hardware-compatibility and generalization. The paper will be stronger if it can put more efforts on addressing these issues.

**Strengths And Weaknesses:**

# Strengths

- The approach is novel, leveraging dynamic state merging to enhance performance.
- The paper is clearly written, with thorough explanations of the method and experiments.
- Experiments are comprehensive, covering language modeling, long-context understanding, and efficiency analysis.

# Weaknesses

- Training Details: The paper focuses on inference, but lacks clarity on training procedures. Dynamic methods often face a gap between training and inference; the paper should address how this is handled.
- Parallelization: While linear attention allows simple decoding, parallel prefilling is more challenging. The paper does not discuss how the two modules of DLA affect model parallelization.

---

> ### Author Rebuttal · Authors · 2026-03-30
>
> **[W1Q1 Training Details]:** We thank the reviewer for raising the important concern regarding training–inference consistency. We clarify that DLA is explicitly designed to minimize this gap rather than rely on heuristic or mismatched policies. Specifically, as described in Sec. 3.1 , DLA adopts a unified criterion (State Information Score) for both training and inference:
> - Training phase: We use a differentiable soft gating mechanism to learn the state partition policy end-to-end. This allows the model to adaptively learn when to merge or split states based on representation variation, instead of relying on fixed rules.
> - Inference phase: We replace soft gating with a hard decision rule based on the same information score, yielding a discrete and efficient state construction procedure.
>
> The inference-time rule is a direct discretization of the learned soft gating, rather than a separate heuristic. Therefore, the objective remains consistent across training and inference, with the only difference being the implementation form (continuous vs. discrete) for efficiency. Compared to prior dynamic methods that rely on heuristic thresholds or non-differentiable segmentation policies, DLA learns the segmentation behavior during training, which substantially reduces the training–inference discrepancy.
>
> **[W2Q2 Parallelization]:** We thank the reviewer for raising this important question. We clarify that DLA largely preserves the parallelization pattern of linear attention, with only a minor and well-structured overhead during prefilling.
>
> - Information-Aware Dynamic State Merging: During prefilling, this module introduces a lightweight sequential dependency within each chunk, since the state information score depends on the current running state and boundary decisions may reset it. As a result, DLA follows the standard chunked parallelization scheme used in linear attention: Specifically, the computation is fully parallel across chunks. Within a chunk, it only requires parallel token-wise computation with a lightweight prefix-style scan for state updates. This scan is local (within chunk) and linear-time, similar to prefix-scan operations already used in efficient linear attention implementations, and does not introduce global synchronization. Therefore, DLA retains high parallel efficiency in prefilling with only modest overhead.
>
> - Capacity-Bounded Memory Modeling: This module operates on a fixed number of states (K) and does not introduce token-level dependencies. All operations are performed at the state level, making it naturally compatible with batching and model parallelism. In fact, the bounded and structured memory layout improves predictability for system-level scheduling.
>
> Overall, DLA maintains the standard chunk-parallel + local-scan paradigm of linear attention. We will clarify these implementation details in the revision.

---

> > ### Author Rebuttal · Reviewer_xBhG · 2026-04-03
> >
> > Thanks for your rebuttal. Your response has addressed my concerns. I wil raise my score

---

### Official Review · Reviewer_JxLi · 2026-03-22

**Soundness:** 3
**Presentation:** 3
**Significance:** 3
**Originality:** 3
**Overall Recommendation:** 4
**Confidence:** 3

**Summary:**

This paper addresses the limitations of existing multi-state linear attention methods in long-context modeling, particularly the use of fixed memory merging policies that fail to adapt to varying token importance. It proposes Dynamic Linear Attention (DLA), which introduces information-aware state merging and a capacity-bounded memory mechanism to better preserve salient information while maintaining efficient computation.

**Compliance With Llm Reviewing Policy:**

Affirmed.

**Final Justification:**

I have no more questions and maintain my score.

**Key Questions For Authors:**

NA

**Strengths And Weaknesses:**

Strengths:

The paper presents a well-motivated improvement over existing multi-state linear attention by explicitly addressing the limitations of fixed merging policies, and the proposed dynamic, information-aware mechanism is both intuitive and easy to integrate into existing architectures. Empirically, the method demonstrates consistent gains across a wide range of tasks, including challenging long-context benchmarks, while also maintaining favorable efficiency in terms of memory and throughput.

Weaknesses:
* The proposed approach fundamentally remains a form of information compression; although dynamic merging mitigates issues of fixed policies, it does not fully resolve the inherent capacity limitations of linear attention in very long-context settings.

* The dynamic partitioning mechanism relies on threshold-based decisions (e.g., the boundary parameter), introducing additional hyperparameters and raising concerns about sensitivity and robustness across tasks and model scales.

* The method introduces extra state management and merging operations, leading to increased implementation complexity compared to standard linear attention, which may hinder practical deployment and efficiency.

* The experimental evaluation does not include realistic long-form generation scenarios (e.g., long document writing), leaving it unclear whether the observed improvements translate to practical generative applications.

---

> ### Author Rebuttal · Authors · 2026-03-31
>
> We thank the reviewer for the insightful comments and address each concern below.
>
> **[W1 On information compression and capacity limitations]:** We agree that DLA, as a form of linear attention, still operates under a compressed memory representation and does not fundamentally remove the capacity bound of linear attention. However, our goal is orthogonal: rather than increasing capacity, DLA optimizes how limited capacity is allocated. As shown in Sec. 3.1, fixed policies incur large deviation when heterogeneous tokens are merged, while DLA adaptively aligns state boundaries with semantic transitions. Therefore, DLA reduces information loss under the same capacity budget, which is precisely the dominant source of degradation in long-context settings.
>
> **[W2 On threshold sensitivity and robustness]:** While inference uses a threshold (τ), it is not a hand-designed heuristic. The decision rule is derived from a learned, differentiable soft gating mechanism during training, and the inference rule is a direct discretization of this learned criterion. As a result, τ operates on a normalized information score and exhibits stable behavior across tasks and scales in our experiments. We will further clarify this point and include sensitivity analysis in the revision.
>
> **[W3 On implementation complexity and deployment]:** DLA introduces additional state management, but this overhead is structured and bounded. In particular, (i) state construction is a lightweight prefix-style scan already common in efficient linear attention implementations, and (ii) memory is explicitly bounded to K states, ensuring predictable layout and cost. In practice, these designs are compatible with existing chunked execution and batching systems, and do not introduce irregular or unbounded overhead.
>
> **[W4 Long-form generation tasks]:** We thank the reviewer for this important question. We would like to clarify that our evaluation already includes realistic long-form generation scenarios. In particular, as described in Table 3 , we evaluate DLA on LongBench summarization tasks, where the model is required to process long documents and generate extended outputs (up to 1K tokens). This setup closely matches practical long-form generation applications such as document summarization and long-form writing, involving both long-context understanding and sustained generation. As shown, DLA outperforms other baseline methods, indicating its strong ability on long-generation tasks.

---

> > ### Author Rebuttal · Reviewer_JxLi · 2026-04-03
> >
> > Thanks for your rebuttal. I hope the authors can clearly state the capacity limitations of linear attention in their revised manuscript.

---

### Decision · Program_Chairs · 2026-04-30

**Decision:**

Accept (regular)

**Comment:**

All reviewers provided a positive overall assessment of the paper, particularly highlighting its novelty and strong empirical results. At the same time, they raised several questions regarding sensitivity to hyperparameters and the breadth of the baseline comparisons. Based on the rebuttal, I believe that most of these concerns have been adequately addressed. Therefore, I am inclined to recommend acceptance.